# Cortical Diffusivity, a Biomarker for Early Neuronal Damage, Is Associated with Amyloid-β Deposition: A Pilot Study

**DOI:** 10.3390/cells14030155

**Published:** 2025-01-21

**Authors:** Justine Debatisse, Fangda Leng, Azhaar Ashraf, Paul Edison

**Affiliations:** 1Division of Neurology, Department of Brain Sciences, Faculty of Medicine, Imperial College London, London W12 0NN, UK; j.debatisse@imperial.ac.uk (J.D.); f.leng18@imperial.ac.uk (F.L.); azhaar.ashraf@imperial.ac.uk (A.A.); 2School of Medicine, Cardiff University, Wales CF14 4YS, UK

**Keywords:** alzheimer’s disease, amyloid-β, cortical mean diffusivity, mild cognitive impairment, tau

## Abstract

Pathological alterations in Alzheimer’s disease (AD) begin several years prior to symptom onset. Cortical mean diffusivity (cMD) may be used as a measure of early grey matter damage in AD as it reflects the breakdown of microstructural barriers preceding volumetric changes and affecting cognitive function. We investigated cMD changes early in the disease trajectory and evaluated the influence of amyloid-β (Aβ) and tau deposition. In this cross-sectional study, we analysed multimodal PET, DTI, and MRI data of 87 participants, and stratified them into Aβ-negative and -positive, cognitively normal, mildly cognitively impaired, and AD patients. cMD was significantly increased in Aβ-positive MCI and AD compared with CN in the frontal, parietal, temporal cortex, hippocampus, and medial temporal lobe. cMD was significantly correlated with cortical thickness only in patients without Aβ deposition but not in Aβ-positive patients. Our results suggest that cMD is an early marker of neuronal damage since it is observed simultaneously with Aβ deposition and is correlated with cortical thickness only in subjects without Aβ deposition. cMD changes may be driven by Aβ but not tau, suggesting that direct Aβ toxicity or associated inflammation causes damage to neurons. cMD may provide information about early microstructural changes before macrostructural changes.

## 1. Introduction

Pathological alterations in Alzheimer’s disease (AD) begin several years before the onset of symptoms [1,2]. During this period, the accumulation of toxic proteins, such as amyloid-β (Aβ) and tau, alongside neuroinflammation, synaptic dysfunction, and other pathological processes, leads to neuronal damage. Although Aβ and tau play crucial roles in AD pathophysiology, how these two critical pathological proteins interact and influence neurodegenerative processes and cortical atrophy remains to be deciphered [2,3,4,5].

The identification of pre-symptomatic patients can help track the pathology, assist in the differentiation of normal ageing from dementia, and enable intervention in the pre-symptomatic phase of AD [6,7,8]. In the last decade, the development and validation of Positron Emission Tomography (PET) imaging biomarkers has enabled the early detection of Aβ, tau deposition, and cerebral glucose metabolism in AD trajectory. In addition to PET imaging, several magnetic resonance imaging (MRI) approaches have been used to quantify hippocampal and regional cortical volume loss [9], cortical thickness [6,10], and microstructural abnormalities in the early stages of neurodegeneration using diffusion-weighted MRI techniques [11,12,13,14,15,16,17,18,19,20,21].

Diffusion tensor imaging (DTI) of grey matter in AD has been used in several studies [21]. Mean diffusivity (MD) is the most widely used metric to assess the average degree of diffusion in all directions, as opposed to fractal anisotropy (FA), which quantifies the directionality of diffusion [21,22,23]. FA is mostly used for assessing white matter fibre tract integrity. However, cortical mean diffusivity (cMD) can be used to assess microstructural alterations in the brain’s grey matter [24,25,26]. Increased cMD in grey matter is a reflection of the breakdown of microstructural barriers to diffusion, which precede volumetric changes [21]. Increased hippocampal cMD and whole brain grey matter cMD have been reported in patients with mild cognitive impairment (MCI) compared with those who are cognitively normal (CN) [20,27]. Increased cMD is demonstrated in patients with MCI who convert to AD compared with patients with MCI who remain stable [17,27,28]. These findings highlight the importance of using grey matter cMD as a useful measure of early grey matter damage in AD.

Several diffusion imaging studies have predominantly focused on white matter changes. Tau has been associated with white matter integrity loss and multiple cognitive functions in AD [29]. Pathological studies suggest that grey matter changes occur prior to white matter changes [21]. Since Aβ and tau have differential spatial patterns, they likely have varying association with grey matter cMD across the AD continuum [30,31]—in individuals that are CN and patients with MCI and AD.

We hypothesised that cortical grey matter cMD is associated with the onset of Aβ deposition—with early and late stages of AD (MCI to AD) serving as a surrogate to the duration of Aβ deposition in this pilot study. We predict that grey matter cMD will be correlated with structural imaging biomarkers, including cortical thickness with MD providing information independent of cortical thickness. The aim of this project was to investigate if we can detect changes in cMD early in the disease trajectory, and to evaluate the impact of Aβ deposition and tau aggregation, and changes in cortical thickness in CN, MCI, and AD.

## 2. Materials and Methods

### 2.1. Subjects

The study was approved by the London Riverside Research Ethics Committee and the National Health Research Services, Health Research Authority, UK (V105/07/2010). The Administration of Radioactive Substances Advisory Committee (ARSAC) gave its approval for the administration of PET tracers. Written informed consent was obtained from all subjects.

The inclusion criteria of the study were as follows: (1) diagnosis of mild cognitive impairment (MCI) was made by a specialist consultant at memory clinics, and the study investigators reviewed the patient according to the Petersen and National Institute of Aging and Alzheimer’s Association (NIA-AA) criteria or Alzheimer’s disease (AD) according to NIA-AA or normal cognitive function for healthy volunteers [32]. Objective memory loss was measured by education-adjusted scores on the Wechsler Memory Scale—Logical Memory; (2) aged between 50 and 85 years; (3) Mini-Mental State Examination (MMSE) score ≥ 28 was considered normal for CN subjects, ≥ 24 for MCI, and ≥ 15 for AD patients. MMSE is a valuable tool to evaluate mental status, which comprises an 11-question test of five areas of cognitive function: orientation, registration, attention and calculation, recall, and language. The maximum score is 30. The MMSE can be administered and is practical to be repeated routinely as it is just 5–10 min long. The instrument is based mainly on verbal response and reading and writing.

Candidates with the following conditions were excluded: (1) major depression, schizophrenia, or schizoaffective disorders; (2) history or signs of other neurological diseases; (3) malignancy within the last 5 years; (4) contraindications for MRI scanning.

In total, 87 subjects were recruited into the study: 50 MCI, 16 AD, and 21 CN subjects. Patients were recruited from the Imperial College memory clinics, dementia registry, and memory clinics around London. All subjects had detailed clinical, neurological, and neuropsychometric evaluation. All participants then underwent MRI scanning and had [^18^F]flutemetamol PET. Meanwhile, 14 MCI, 10 AD, and 6 CN had [^18^F]AV1451 PET scans.

### 2.2. Image Acquisition

#### 2.2.1. MRI

MRI scans were acquired with a 3 Tesla Siemens Verio scanner using a 32-channel head coil. T1-weighted magnetisation-prepared rapid gradient echo sequence (MPRAGE) images were acquired with TR = 2300 ms, TE = 2.98 ms, FA = 9°, TI = 900 ms, 1 × 1 × 1 mm^3^ voxel, anteroposterior phase encoding, and FOV = 256 × 256 mm^2^. Diffusion tensor images were acquired in 64 diffusion directions as 62 axial slices using an EPI sequence TR = 9000 ms, TE = 99 ms, anteroposterior phase encoding, 2 × 2 × 2 mm^3^ voxel, FOV = 256 × 256 mm^2^, bandwidth = 1562 Hz/Px, and echo spacing = 0.72 ms.

#### 2.2.2. PET

[^18^F]flutemetamol was manufactured by GE Healthcare, Amersham, UK. Scans were performed at the Imperial College Clinical Imaging Facility with a Siemens Biograph 6 scanner. [^18^F]flutemetamol at a dose of 183.4 (±5.3) MBq was injected intravenously in 8 mL saline followed by a 10 mL saline flush. Data were acquired in 3D list mode from 90 to 120 min following injection (6 × 5 min frames). Image reconstruction was performed by filtered back projection with an attenuation correction. Post-reconstruction 5 mm Gaussian smoothing was performed.

[^18^F]AV1451 was manufactured by Imanova/Invicro, and the scans were performed on a Siemens Biograph 6 scanner. A target dose of 180 MBq [^18^F]AV1451 was injected intravenously in 8 mL saline followed by a 10 mL saline flush. Data were acquired in 3D list mode from 0 to 120 min. Image reconstruction was performed by filtered back projection with an attenuation correction. Post-reconstruction 5 mm Gaussian smoothing was performed.

### 2.3. Image Processing

#### 2.3.1. MRI

Regional cortical thickness was measured on T1 images using FreeSurfer v.6.0 (Harvard Medical School; http://surfer.nmr.mgh.harvard.edu/ (accessed on 16 January 2025)). Briefly, Freesurfer’s method uses intensity and continuity information from the T1-weighted MR volume to generate cortical thickness representations, which are estimated as the distance between the grey/white and grey/CSF boundaries [33].

Diffusion tensor images were denoised, motion-, distortion-, and eddy-current-corrected, and brain-extracted using FSL software (FMRIB Software Library, v6.0) with the FSL FDT package. Skull and non-brain tissue were removed, and then the tensor model was fitted to calculate the cMD maps for each participant individually. The tensor model was used to calculate diffusion in the principal direction and the two perpendicular directions. cMD was obtained by taking the sum of the diffusion in the principal direction and diffusion in the two perpendicular directions and dividing it by three. cMD images were co-registered to each subjects’ anatomical scan. The resulting images were then spatially normalised and co-registered to the MNI space using FSL v6.0 [34].

#### 2.3.2. PET

PET image processing was performed using Statistical Parametric Mapping 12 (SPM12, Wellcome Trust Centre for Neuroimaging, UCL, London, UK). PET images were co-registered to their T1 MRI and transformed into the Montreal Neurological Institute (MNI) space. We calculated a 90–120 min standard uptake value ratio (SUVR) of [^18^F]flutemetamol using cerebellum grey matter as the reference region, as previously validated [35].

Using SPM12, an individualised object map was created for each participant using the following steps: (1) Individual MRI was segmented into grey matter, white matter, and CSF. The binarised grey matter mask was created using a threshold of 0.5; (2) the binarised individual grey matter mask was then applied to the probabilistic region of interest (ROI) using Hammers’ atlas in MNI space [36] to create an individualised object map. Regional imaging parameters were estimated for frontal, temporal, parietal, and occipital cortical regions. We further evaluated hippocampus and medial temporal lobes separately.

### 2.4. Determining the Aβ Status

Based on ROI analysis of the [^18^F]flutemetamol SUVR image, subjects were classified as Aβ-positive or Aβ-negative. Since we were interested in evaluating early changes in Aβ deposition and cortical diffusivity, we wanted to identify patients with minimal Aβ deposition before Aβ levels reached the Aβ positivity cut off described in the literature; subjects were determined as Aβ-positive if there were one more or cortical regions (frontal, parietal, temporal, occipital lobe) with binding greater than the CN mean + 2 standard deviations as previously published [35,37]. Using “CN mean + 2 standard deviations” as the cut off for positivity, we had 43 subjects who were Aβ-positive. This number would have been reduced to 29, if we were to use the [^18^F]flutemetamol SUVR threshold of 1.57 as the Aβ positivity cut off [35].

### 2.5. Statistical Analysis

Statistical analyses of numeric variables were performed using GraphPad Prism version 9.2.0 for Windows, GraphPad Software, San Diego, CA, USA, www.graphpad.com (accessed on 16 January 2025). *p*-values were calculated by one-way ANOVA and the Tukey post hoc test. The imaging parameters were correlated using Spearman’s rho correlation coefficient. A *p*-value < 0.05 was considered significant.

## 3. Results

### 3.1. Demographic Characteristics

We removed Aβ-positive CN (*n* = 4) and Aβ-negative AD (*n* = 3) from further analysis (Figure 1). Of the remaining 80 patients, there were 17 CN, 26 Aβ-negative MCI, 24 Aβ-positive MCI, and 13 AD.

The demographic and clinical characteristics of the diagnostic groups are summarised in Table 1. Mean ages (±SD) were higher in Aβ-positive MCI (73.9 ± 7.3, *p* = 0.003) and AD (75.0 ± 5.7, *p* = 0.005) than those of the CN clinical diagnostic group (62.9 ± 8.4). Mean MMSE scores were significantly lower in the AD group (24 ± 4; *p* < 0.001) than in the three other clinical diagnostic groups (i.e., 29 ± 2 for CN; 28 ± 2 for Aβ-negative MCI and 27 ± 2 for Aβ-positive MCI). There were no group differences in education between the three different groups.

### 3.2. cMD Changes Are Observed Simultaneously as Aβ Deposition

Individual representations of cMD, Aβ and tau deposition, and cortical thickness are illustrated in Figure 2 for the frontal cortex (Figure 2A), parietal cortex (Figure 2B), temporal cortex (Figure 2C), and regional cortical thickness (Figure 2D).

For each region of interest, i.e., the frontal, parietal, temporal cortex, hippocampus, and medial temporal lobe, a one-way ANOVA was performed to compare the means of the four different groups, and cMD was significantly different among the groups: F (3, 71) = 5.84, *p* = 0.001 in the frontal cortex; F (3, 71) = 5.05, *p* = 0.003 in the temporal cortex; F (3, 71) = 5.60; *p* = 0.002 in the parietal cortex; F (3, 71) = 4.25, *p* = 0.008 in the hippocampus; F (3, 71) = 3.40, *p* = 0.022 in the medial temporal lobe (Table 2). Tukey post hoc tests revealed that, compared to CN, cMD was significantly increased in AD subjects in all tested regions, and cMD was significantly increased in Aβ-positive MCI compared with CN in the frontal cortex, parietal cortex, and hippocampus (Table 2, Figure 2A–C).

As expected, Aβ deposition characterised by [^18^F]Flutemetamol SUVR was also significantly different among the groups, and Tukey post hoc tests revealed that, compared with CN, Aβ deposition was significantly increased in AD and Aβ-positive MCI subjects in the frontal cortex, temporal cortex, and parietal cortex (*p* < 0.001 for all, Table 2, Figure 2A–C).

We also found that tau deposition, characterised by [^18^F]AV1451 SUVR, was significantly different among groups in the temporal cortex (F (3, 26) = 7.605, *p* = 0.001), parietal cortex (F (3, 26) = 3.367, *p* = 0.034), hippocampus (F (3, 26) = 8.187, *p* = 0.001), and MTL (F (3, 26) = 10.15, *p* < 0.001, Table 2, Figure 2A–C). Tukey post hoc tests revealed that, compared with CN, tau deposition was significantly increased in AD subjects in the temporal cortex (*p* = 0.003), hippocampus (*p* = 0.008), and MTL (*p* = 0.002), and tau deposition was also significantly increased in Aβ-positive MCI compared with CN in MTL (*p* = 0.041, Table 2, Figure 2A–C).

Lastly, we also found that cortical thickness was significantly different among groups in the temporal cortex (F (3, 76) = 10.08, *p* < 0.001), parietal cortex (F (3, 76) = 14.80, *p* < 0.001), and in the whole brain-averaged cortical thickness (F (3, 76) = 10.01, *p* < 0.001, Table 2, Figure 2D). Tukey post hoc tests revealed that, compared with CN, cortical thickness was significantly decreased in AD subjects in the temporal cortex (*p* < 0.001), the parietal cortex (*p* < 0.001), and mean cortical thickness (*p* < 0.001, Table 2, Figure 2D).

From this ROI analysis, we found that cMD was significantly increased as early as Aβ deposition in various brain regions, and cMD was not significantly increased in Aβ-negative MCI compared with CN.

### 3.3. cMD Is Significantly Associated with Cortical Atrophy When Other Pathologies Are Not Present

We then investigated the relationship between cortical thickness and cMD in each ROI and each clinical diagnostic group. In the whole cohort, higher levels of cMD were associated with lower cortical thickness in the frontal (rho = −0.30, *p* = 0.010), temporal (rho = −0.41, *p* = <0.001), parietal cortices (rho = −0.36, *p* = 0.002), and hippocampus (rho = −0.36, *p* = 0.002). Higher levels of Aβ were associated with higher cMD in the frontal (rho = 0.30, *p* = 0.010), temporal (rho = 0.28, *p* = 0.015), and parietal cortices (rho = 0.31, *p* = 0.008). No significant association was observed between tau and cMD in the frontal (rho = 0.32, *p* = 0.095), temporal (rho = 0.30, *p* = 0.120), parietal cortices (rho = 0.04, *p* = 0.842), and hippocampus (rho = 0.17, *p* = 0.397).

Based on subgroup analysis (Figure 3), we found a strong negative correlation between cortical thickness and cMD in subjects who were Aβ-negative (i.e., CN and Aβ-negative MCI) in the parietal (Figure 3A, rho = −0.51, *p* < 0.001), temporal cortex (Figure 3C, rho = −0.53, *p* < 0.001), and whole brain (Figure 3E, rho = −0.49, *p* = 0.001). Interestingly, we found no correlation in subjects who were Aβ-positive (i.e., Aβ-positive MCI and AD) in the same regions: parietal cortex (Figure 3B, rho = −0.06, *p* = 0.738), temporal cortex (Figure 3D, rho = −0.22, *p* = 0.211), and whole brain (Figure 3F, rho = −0.0328, *p* = 0.852). Spearman correlation analysis between Aβ and cMD in the Aβ-positive and -negative populations showed no significant correlations (Figure 4). These results suggest that cortical thickness and cMD are only related in the absence of pathologies such as Aβ deposition.

We did not find any significant correlation in subjects who were Aβ-positive (i.e., Aβ-positive MCI and AD) in the same regions: hippocampus (rho = −0.32, *p* = 0.06) and in the medial temporal lobe (rho = −0.28, *p* = 0.102).

In Aβ-negative and Aβ-positive subjects, we did not find any significant correlations between cMD and tau deposition in any of the previously tested brain regions. Spearman correlation analysis between tau deposition and cMD in the Aβ-positive and -negative populations showed no significant correlations (Figure 5). Cortical thickness was also not significantly correlated with tau or Aβ deposition, neither at a regional nor a global level in those Aβ-negative and Aβ-positive subjects.

## 4. Discussion

This pilot study examined the potential relationship between the mean diffusivity of cMD and the initiation of Aβ deposition, considering different stages of AD as a proxy for the duration of Aβ accumulation. We demonstrated that cMD was significantly increased in Aβ-positive MCI and AD compared with CN in the frontal, parietal, temporal cortex, hippocampus, and medial temporal lobe. We also found that cMD was significantly correlated with cortical thickness only in patients without Aβ deposition, and was not significant in Aβ-positive patients. We anticipated a correlation between cMD and structural imaging biomarkers, particularly cortical thickness, as cMD offers unique information independent of cortical thickness. The primary objective of this investigation was to identify early changes in cMD during the disease trajectory and establish its relationship with Aβ and tau deposition, and alterations in cortical thickness across the cognitive continuum encompassing CN, MCI, and AD patients. This enabled us to understand the influence of Aβ, one of the earliest changes in the AD trajectory along with gaining insights into the potential utility of cMD as an early marker of AD pathology and neurodegenerative changes. Our hypotheses were guided by the notion that Aβ deposition may have a deleterious effect on cMD early in the disease trajectory and serve as a sensitive marker of microstructural alterations associated with Aβ deposition. This may provide valuable information complementary to traditional measures like cortical thickness. While Aβ deposition was associated with changes in cMD, no such association was found with tau deposition in this relatively small number of subjects.

Our findings shed light on several important aspects related to cMD and its association with AD pathology: (1) cMD is an early marker of neuronal damage, and it appears in Aβ-positive MCI, and may coincide with the Aβ deposition. Significant increases in cortical cMD were observed in brain regions of symptomatic Aβ-positive MCI and AD participants compared with symptomatic Aβ-negative MCI and CN participants. (2) cMD changes are driven by Aβ but not by tau, which may suggest that direct Aβ toxicity or associated inflammation causes damage to dendrites. (3) cMD is associated with cortical thickness when Aβ is not present (Figure 6). cMD is correlated with cortical thickness only in CN and Aβ-negative MCI participants, suggesting that cMD might be associated with cortical atrophy when other pathologies (such as Aβ deposition) are not present. When other pathologies are present, those pathologies influence the cortical thickness. This suggests that Aβ deposition may have an independent effect on neuronal damage, which further underscores the potential utility of cMD as an early biomarker. Detecting and monitoring disease progression over time can be crucial in the early stages of AD, and cMD holds promise as a valuable tool.

Our findings highlight the significance of cortical cMD as a sensitive indicator of neuronal damage in the context of AD pathology, particularly Aβ deposition. The observed independent effect of Aβ on cMD underscores its potential as an early biomarker for tracking disease progression, providing insights into the underlying mechanisms, and guiding future research and therapeutic interventions.

Our findings are consistent with the Aβ cascade hypothesis which assumes a serial model of causality whereby Aβ initiates a series of events leading to tau hyperphosphorylation and neurodegeneration [38]. We show here that cMD is not significantly affected in individuals who are Aβ-negative MCI. This could imply that prior to the presence of Aβ, the grey matter topology may not be significantly impacted. The accrual of Aβ as observed in Aβ-positive MCI may cause local disruption of the dendrites. This suggests that the integrity of the dendrites is compromised even in the early stages of the AD trajectory, as observed in the present study. These findings demonstrate that grey matter diffusivity is an early finding in the AD trajectory and is influenced by Aβ deposition, and is perhaps associated with neuroinflammation [13,17,19,21]. Changes in grey matter diffusivity is hypothesised to be the consequence of the breakdown of microstructural barriers (intracellular organelles, cell membranes, etc.), associated with and/or amplified by the simultaneous presence of Aβ [21,25,26,39]. There may be an inflammatory state associated with neuronal/glial swelling and inflammatory cell recruitment. This could disrupt the cell membrane, leading to changes in cMD.

cMD was negatively associated with cortical thickness in patients who were Aβ-negative in this study. This suggests that cMD closely reflects disease activity in the period [40] approaching Aβ positivity, since this correlation between cMD and cortical thickness was lost in patients who were Aβ-positive. In agreement with our findings, a study by Weston et al. (2020) found that most significant cortical MD changes are observed in the pre-symptomatic familial AD phase, but disappear during the symptomatic phases [40]. This suggests that MD indicates disease activity in the period approaching the onset of clinical symptoms [41,42,43,44]. A study in AD showed different clinical diagnostic patterns (CN, MCI, AD) in the association between whole brain cMD and cortical volumes. However, their Aβ status was not assessed [27]. Our pilot study demonstrates that cMD might be a valuable biomarker of neurodegeneration reflecting the upstream microstructural changes happening prior to macrostructural changes like atrophy. Multimodal and longitudinal imaging studies on large datasets would be needed to evaluate these propositions.

We found that cortical thickness was significantly lower in AD participants compared with the three other cognitive diagnostic groups. Interestingly, we also found that cortical thickness was strongly correlated with cMD in CN and Aβ-negative MCI, with increasing cMD being associated with decreasing cortical thickness. In the absence of other pathologies, increased cMD may lead to neuronal damage and a subsequent reduction in cortical thickness. In the presence of Aβ and tau deposition, these pathological processes may influence cMD and contribute to a subsequent reduction in cortical thickness. Measurement of cortical thickness using the FreeSurfer package is a common and well-validated method of assessing macrostructural cortical change [40,41]. The relationship between thickness and MD thus provides further face validity for cortical MD as a marker of cortical integrity/degeneration, with both decreased thickness and increasing MD likely constituting part of the same pathological continuum. The presence of an association among healthy ageing non-carriers also supports this theory, indicating that both cortical MD and cortical thickness are measuring an underlying metric, i.e., neuronal integrity/degeneration (albeit differing factors and/or time points). These measurements continually change in a progressive manner, albeit to a milder degree in both neurodegenerative disease and healthy ageing. Changes in microstructural measures like MD may be upstream to macrostructural changes in thickness, which may provide an earlier measure change. However, this will need confirmation through further studies, preferably with longitudinal assessment.

We did not find any increase in either cMD or cortical thickness, on the ROI analysis, in Aβ-negative MCI participants compared with CN participants. Even if we assume that neuronal death occurs early in the course of the disease, several mechanisms may explain the absence of increased cMD in Aβ-negative MCI participants: (1) there could be an inflammatory state associated with neuronal/glial swelling and inflammatory cell recruitment that could lead to a decreased cMD. Some evidence suggests a biphasic evolution of cMD, with an initial decrease in the early stages of the disease, followed by an increase in cMD [39]. This may be consistent with the early and late peaks of microglial activation described in AD [45]. There is strong evidence supporting this hypothesis, with several studies showing the effect of cell hypertrophy, glial recruitment, and activation, thus modifying the diffusion properties of water molecules and adding new barriers to induce a decrease in diffusivity in those areas [46,47]. Further studies with simultaneous inflammation, Aβ, tau, and diffusion imaging in the early stages of dementia would be of particular interest for identifying the biological processes involved in diffusivity changes.

There are technological considerations for the non-significant changes observed in Aβ-negative MCI compared with CN. The present study used PET imaging, which detects fibrillar forms of Aβ but not soluble oligomers. It has been suggested that soluble Aβ oligomers, readily measured by measuring CSF acquired through lumbar puncture, precedes PET measurements of Aβ fibrils [48,49]. The oligomeric forms of Aβ have been considered the most toxic and pathogenic [50]. Aβ42 oligomers isolated from typical late-onset AD brains decrease synapse density, inhibit LTP, and enhance long-term synaptic depression in rodent hippocampus, and their intraventricular injection impairs memory in healthy adult rats [50]. Moreover, human Aβ42 oligomers induce tau hyperphosphorylation at AD-relevant epitopes and cause neuritic dystrophy in cultured rat neurons; co-administering Aβ antibodies fully prevents this [50]. Future studies should elucidate the relationship between Aβ oligomers and their effects on diffusivity parameters in Aβ-negative MCI.

Our results demonstrate that Aβ and tau have distinct accumulation patterns but overlap with changes in cMD in various brain regions. In the prodromal stage of AD dementia, Aβ accumulation was found to be significantly increased in the frontal, temporal, and parietal regions, but not in the medial temporal regions, including the hippocampus, in keeping with sites of early Aβ accumulation [51]. Meanwhile, tau deposition was confined to the medial temporal cortex at the prodromal stage but spread to the frontotemporal regions (higher-order association areas) in patients with AD [52]. Our preliminary observations suggest that the changes in cMD overlap with Aβ and tau, with cMD changes preceding tau deposition. For example, tau accumulation occurred in the frontotemporal regions at AD dementia stage, while cMD changes were observed at the stage of MCI in patients who were Aβ-positive as well as in patients with AD.

One of the limitations of our study was the relatively low number of subjects in each diagnostic group. Nonetheless, we had well characterised 87 subjects with data acquired on the same scanner with the same acquisition parameters, thus strengthening the reliability of measures, but limiting the number of subjects. To analyse and compare a substantial number of subjects data, multicentric studies are needed with the need for standardisation and harmonisation of imaging protocols, crucial for attaining reliable and comparable data. DTI has a relatively poor imaging resolution compared with anatomical structural imaging, and the CSF contribution in the cMD signal is important to consider [24,53]. This study has demonstrated for the first time that in the absence of Aβ deposition, cMD is associated with cortical thickness, while Aβ and tau may have significant influences on cortical thickness. While this preliminary cross-sectional study suggests that microstructural changes occur due to Aβ toxicity, prospective longitudinal studies are required with repetitive imaging and blood sampling to validate the present findings. Moreover, the effects of comorbidities, including hypertension, diabetes, and hypercholesteremia, on cMDs in AD should be thoroughly investigated.

The annual rates of 0.5% for global atrophy have been reported in healthy ageing [54]. The annual rates for global atrophy are more pronounced in neurodegenerative diseases, i.e., 2.4% in patients with AD, 3.2% in patients with FTD, and 1.4% in patients with DLB [55,56]. Further longitudinal studies are required to determine the threshold needed to distinguish between regular age-related changes in cortical atrophy in AD and other neurodegenerative diseases, and to determine their relationship with cMD signal alongside Aβ and tau pathologies.

## 5. Conclusions

We investigated the cortical changes using structural MRI, DTI, and PET imaging markers. We demonstrated that cMD is increased simultaneously with Aβ deposition. In subjects without Aβ pathology (i.e., CN and Aβ-negative MCI), we found a strong association of cMD with cortical thickness. This association between cMD and cortical thickness was not found in the later stages of the disease (i.e., Aβ-positive MCI and AD). Taken together, cMD provides valuable information about early microstructural changes before macrostructural changes.

## Figures and Tables

**Figure 1 cells-14-00155-f001:**
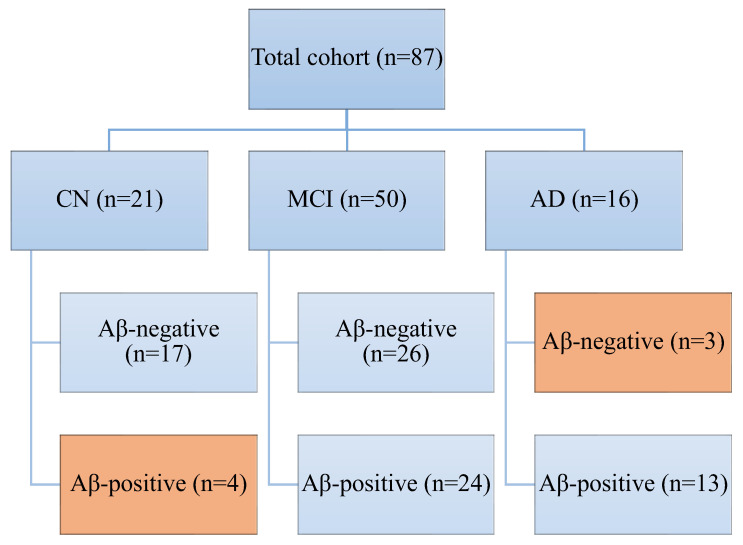
Flow diagram demonstrating patients screening and inclusions. CN Aβ-positive and Aβ-negative AD were excluded from further analysis. AD—Alzheimer’s disease; CN—cognitively normal; MCI—mild cognitive impairment.

**Figure 2 cells-14-00155-f002:**
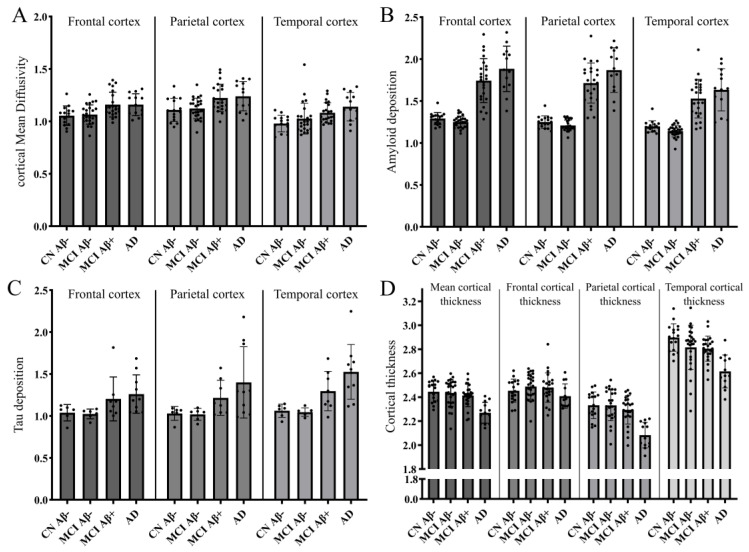
Bar charts representing cMD (**A**); Aβ deposition ([^18^F]flutemetamol SUVR; (**B**) and tau deposition ([^18^F]AV1451 SUVR; (**C**) in frontal, parietal, and temporal cortex. Line and error bars represent mean and 95% confidence interval. Aβ and tau are expressed as SUVR. cMD is expressed as 0.103 mm^2^/s. Bar charts (**D**) of clinical diagnostic group comparison of mean, frontal, parietal, and temporal cortical thickness.

**Figure 3 cells-14-00155-f003:**
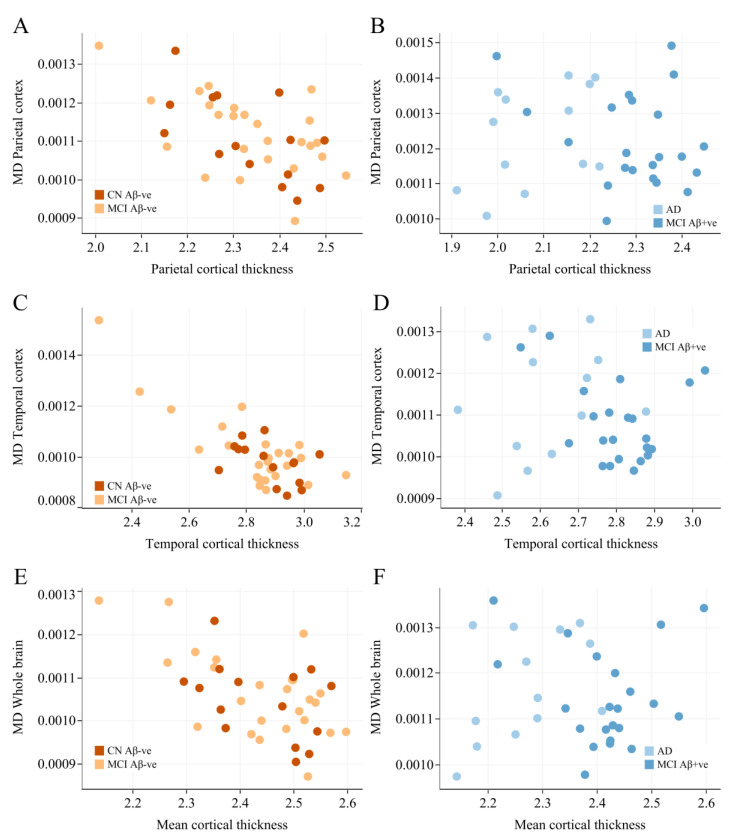
Relationship between cortical thickness and mean diffusivity in the Aβ-negative (CN and Aβ-negative MCI) and Aβ-positive subjects (Aβ-positive MCI and AD). Spearman’s rank correlation are used to determine r and *p*-values in the parietal cortex ((**A**) rho = −0.51, *p* < 0.001; (**B**) rho = −0.06, *p* = 0.738), temporal cortex ((**C**) rho = −0.53, *p* < 0.001; (**D**) rho = −0.22, *p* = 0.211), and whole brain ((**E**) rho = −0.0328, *p* = 0.852; (**F**) rho = −0.0328, *p* = 0.852). AD—Alzheimer’s disease; CN—cognitively normal; MCI—mild cognitive impairment; cMD—mean diffusivity. Figure 3A, rho = −0.51, *p* < 0.001), temporal cortex (Figure 3C, rho = −0.53, *p* < 0.001) and whole brain (Figure 3E, rho = −0.49, *p* = 0.001). Interestingly, we found no correlation in subjects who were Aβ-positive (i.e., Aβ-positive MCI and AD) in the same regions: parietal (Figure 3B, rho = −0.06, *p* = 0.738), temporal cortex (Figure 3D, rho = −0.22, *p* = 0.211), and whole brain (Figure 3F, rho = −0.0328, *p* = 0.852).

**Figure 4 cells-14-00155-f004:**
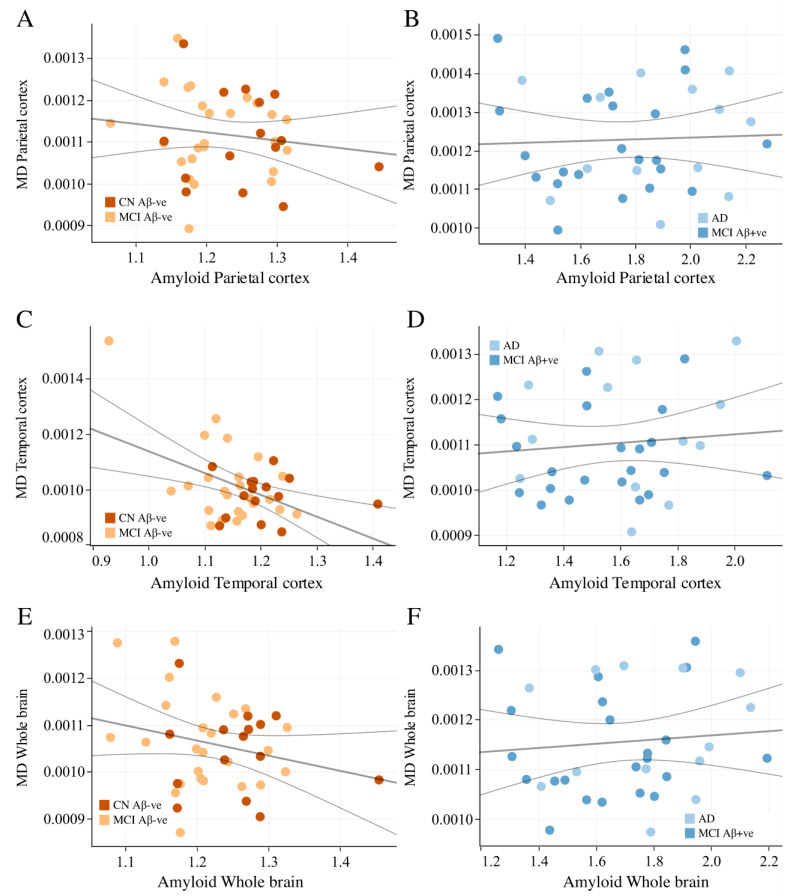
Relationship between amyloid deposition and mean diffusivity in the Aβ-negative (CN and Aβ-negative MCI) and Aβ-positive subjects (Aβ-positive MCI and AD). Spearman’s rank correlation are used to determine r and *p*-values in parietal cortex ((**A**) rho = −0.125, *p* = 0.440; (**B**) rho = 0.087, *p* = 0.616), temporal cortex ((**C**) rho = −0.151, *p* = 0.350; (**D**) rho = 0.078, *p* = 0.654), and whole brain ((**E**) rho = −0.135, *p* = 0.405; (**F**) rho = 0.132, *p* = 0.447). No correlation was observed between Aβ and cMD. AD—Alzheimer’s disease; CN—cognitively normal; MCI—mild cognitive impairment; cMD—mean diffusivity.

**Figure 5 cells-14-00155-f005:**
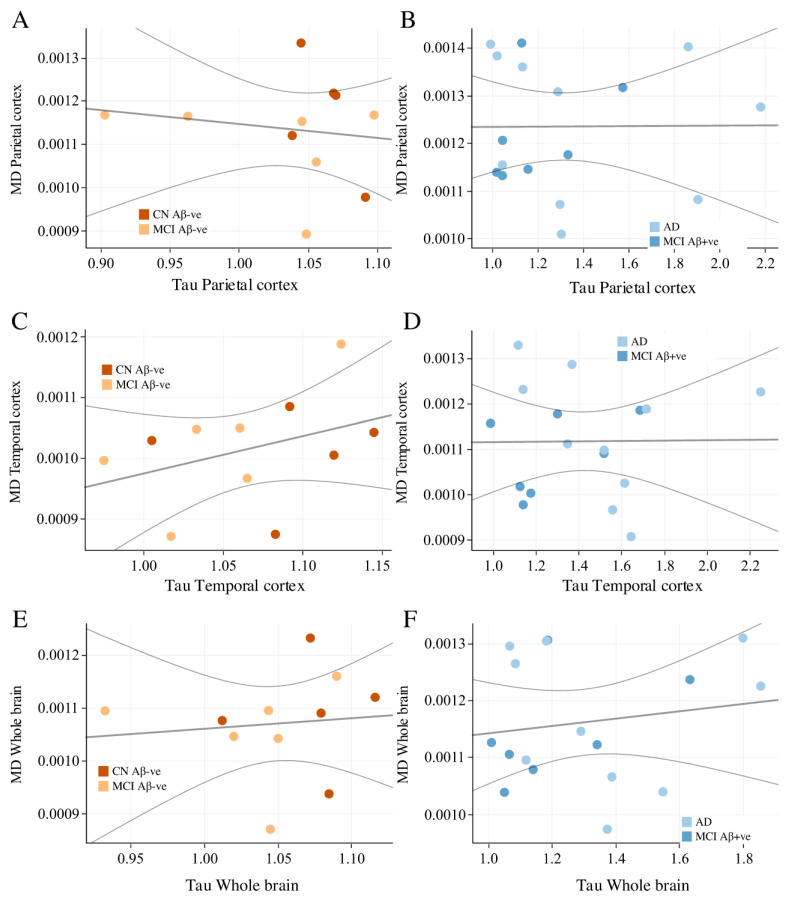
Relationship between tau deposition and mean diffusivity in the Aβ-negative (CN and Aβ-negative MCI) and Aβ-positive subjects (Aβ-positive MCI and AD). Spearman’s rank correlation are used to determine r and *p*-values in parietal cortex ((**A**) rho = −0.091, *p* = 0.797; (**B**) rho = −0.174, *p* = 0.503), temporal cortex ((**C**) rho = 0.400, *p* = 0.225; (**D**) rho = −0.064, *p* = 0.809), and whole brain ((**E**) rho = 0.273, *p* = 0.418; (**F**) rho = 0.113, *p* = 0.666). No correlation was observed between tau and cMD. AD—Alzheimer’s disease; CN—cognitively normal; MCI—mild cognitive impairment; cMD—mean diffusivity.

**Figure 6 cells-14-00155-f006:**
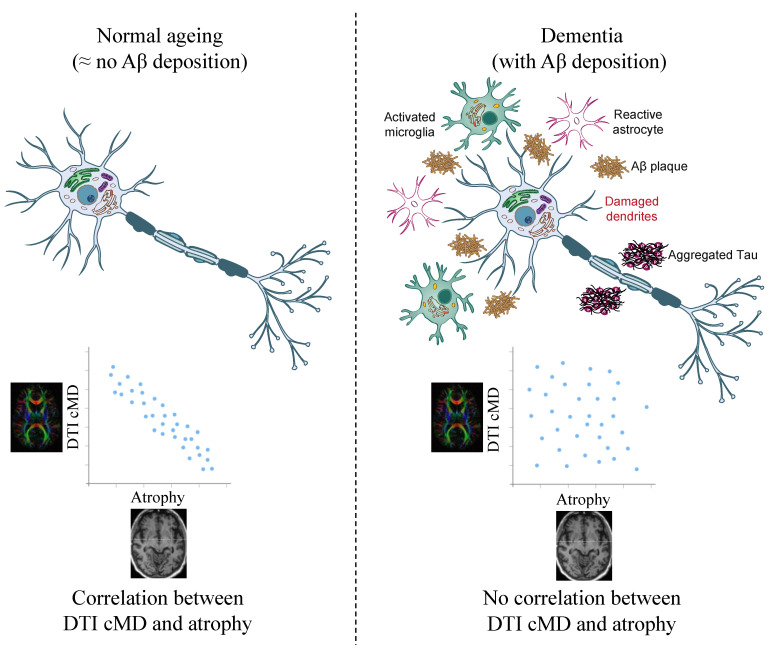
Hypothetical framework of pathological events leading to neuronal damage in normal ageing and dementia. cMD is correlated with cortical thickness only in CN and Aβ-negative MCI participants. This suggests that cMD is associated with cortical atrophy when other pathologies (such as Aβ deposition) are not present. When other pathologies are present, such as neuroinflammation and tau aggregation, those pathologies induce damage to the dendrites and influence the cortical thickness.

**Table 1 cells-14-00155-t001:** Demographics and clinical characteristics according to the clinical diagnostic group.

	CN	Aβ-Negative MCI		Aβ-Positive MCI		AD		One-Way ANOVA	
	Mean (SD)	Mean (SD)	Tukey Post Hoc Test vs. CN	Mean (SD)	Tukey Post Hoc Test vs. CN	Mean (SD)	Tukey Post Hoc Test vs. CN	F	*p*-Value
Age, y (SD)	62.9 (8.4)	67.6 (9.5)	*p* = 0.400	73.9 (7.3)	*p* = 0.003	75.0 (5.7)	*p* = 0.005	F (3, 63) = 6.45	*p* < 0.001
Sex, Female (%)	47	48	*p* = 0.971	40	*p* = 1.000	46	*p* = 1.000	F (3, 74) = 0.12	*p* = 0.951
MMSE (SD)	29 (2)	28 (2)	*p* = 0.540	27 (2)	*p* = 0.421	24 (4)	*p* < 0.001	F (3, 64) = 9.84	*p* < 0.001

Data are expressed as the mean (SD) or percentage. *p*-values were calculated by one-way ANOVA and the Tukey post hoc test. A *p*-value < 0.05 was considered as significant. AD—Alzheimer’s disease; CN—cognitively normal; MCI—mild cognitive impairment; MMSE—Mini-Mental State Examination.

**Table 2 cells-14-00155-t002:** Comparison of mean diffusivity, [^18^F]flutemetamol SUVR, [^18^F]AV1451 SUVR, and cortical thickness values in the ROIs.

		CN	Aβ-Negative MCI		Aβ-Positive MCI		AD		One-Way ANOVA	
Parameter	Brain Region	Mean (SD)	Mean (SD)	Tukey Post Hoc Test vs. CN	Mean (SD)	Tukey Post Hoc Test vs. CN	Mean (SD)	Tukey Post Hoc Test vs. CN	F	*p*-Value
Mean Diffusivity	Frontal cortex	0.00105 (0.00009)	0.00106 (0.00009)	*p* = 0.989	0.00116 (0.00012)	*p* = 0.015	0.00116 (0.00010)	*p* = 0.041	F (3, 71) = 5.84	*p* = 0.001
	Temporal cortex	0.00098 (0.00008)	0.00103 (0.00015)	*p* = 0.628	0.00108 (0.00010)	*p* = 0.057	0.00114 (0.00014)	*p* = 0.004	F (3, 71) = 5.05	*p* = 0.003
	Parietal cortex	0.00111 (0.00011)	0.00113 (0.00011)	*p* = 0.987	0.00122 (0.00013)	*p* = 0.027	0.00124 (0.00014)	*p* = 0.026	F (3, 71) = 5.60	*p* = 0.002
	Hippocampus	0.00090 (0.00008)	0.00105 (0.00032)	*p* = 0.311	0.00112 (0.00018)	*p* = 0.074	0.00125 (0.00039)	*p* = 0.005	F (3, 71) = 4.25	*p* = 0.008
	Medial temporal lobe	0.00091 (0.00007)	0.00102 (0.00025)	*p* = 0.306	0.00106 (0.00012)	*p* = 0.106	0.00113 (0.00024)	*p* = 0.015	F (3, 71) = 3.40	*p* = 0.022
[^18^F]flutemetamol SUVR	Frontal cortex	1.29 (0.07)	1.25 (0.07)	*p* = 0.893	1.74 (0.26)	*p* < 0.001	1.88 (0.27)	*p* < 0.001	F (3, 76) = 54.13	*p* < 0.001
	Temporal cortex	1.20 (0.07)	1.14 (0.07)	*p* = 0.709	1.53 (0.23)	*p* < 0.001	1.63 (0.25)	*p* < 0.001	F (3, 76) = 38.88	*p* < 0.001
	Parietal cortex	1.25 (0.07)	1.21 (0.06)	*p* = 0.884	1.71 (0.24)	*p* < 0.001	1.87 (0.27)	*p* < 0.001	F (3, 76) = 65.56	*p* < 0.001
	Hippocampus	1.42 (0.08)	1.30 (0.15)	*p* = 0.078	1.41 (0.17)	*p* = 1.0	1.39 (0.19)	*p* = 0.982	F (3, 76) = 3.113	*p* = 0.031
	Medial temporal lobe	1.27 (0.07)	1.18 (0.11)	*p* = 0.159	1.32 (0.15)	*p* = 0.544	1.33 (0.19)	*p* = 0.583	F (3, 76) = 6.135	*p* = 0.001
[^18^F]AV1451 SUVR	Frontal cortex	1.04 (0.10)	1.02 (0.06)	*p* = 0.999	1.20 (0.26)	*p* = 0.430	1.26 (0.23)	*p* = 0.158	F (3, 26) = 2.679	*p* = 0.068
	Temporal cortex	1.06 (0.08)	1.05 (0.05)	*p* = 0.999	1.30 (0.23)	*p* = 0.266	1.53 (0.33)	*p* = 0.003	F (3, 26) = 7.605	*p* = 0.001
	Parietal cortex	1.03 (0.08)	1.02 (0.07)	*p* = 1.000	1.22 (0.21)	*p* = 0.604	1.40 (0.42)	*p* = 0.069	F (3, 26) = 3.367	*p* = 0.034
	Hippocampus	1.13 (0.16)	1.07 (0.06)	*p* = 0.952	1.42 (0.21)	*p* = 0.054	1.49 (0.25)	*p* = 0.008	F (3, 26) = 8.187	*p* = 0.001
	Medial temporal lobe	1.06 (0.11)	1.00 (0.06)	*p* = 0.961	1.35 (0.24)	*p* = 0.041	1.46 (0.22)	*p* = 0.002	F (3, 26) = 10.15	*p* < 0.001
Cortical thickness	Frontal cortex	2.45 (0.10)	2.49 (0.11)	*p* = 0.730	2.48 (0.12)	*p* = 0.872	2.41 (0.10)	*p* = 0.670	F (3, 76) = 1.804	*p* = 0.154
	Temporal cortex	2.90 (0.11)	2.82 (0.18)	*p* = 0.261	2.80 (0.11)	*p* = 0.178	2.61 (0.14)	*p* < 0.001	F (3, 76) = 10.08	*p* < 0.001
	Parietal cortex	2.33 (0.11)	2.33 (0.13)	*p* = 1.000	2.29 (0.12)	*p* = 0.747	2.08 (0.11)	*p* < 0.001	F (3, 76) = 14.80	*p* < 0.001
	Mean cortical thickness	2.45 (0.09)	2.43 (0.11)	*p* = 0.986	2.41 (0.09)	*p* = 0.747	2.27 (0.09)	*p* < 0.001	F (3, 76) = 10.01	*p* < 0.001

Data are mean ± SD values, and *p*-values were calculated by one-way ANOVA and the Tukey post hoc test. A *p*-value < 0.05 was considered as significant. AD—Alzheimer’s disease; CN—cognitively normal; MCI—mild cognitive impairment; SUVR—standardised uptake volume ratio.

## Data Availability

The datasets used and/or analysed during the current study are presented in the main manuscript whenever possible. Extra details can be made available from the corresponding author upon reasonable request.

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
