# Peer review of "Cortical Diffusivity, a Biomarker for Early Neuronal Damage, Is Associated with Amyloid-β Deposition: A Pilot Study"

_cells, 2025, doi:10.3390/cells14030155_

Round 1
Reviewer 1 Report
Comments and Suggestions for Authors
This manuscript (cells-3388217) investigated the cortical mean diffusivity (cMD) as an early biomarker for neuronal damage in Alzheimer's disease, and evaluated the influence of Aβ and tau deposition. By analyzing multimodal PET, DTI, and MRI data from 87 participants, the study found that cMD significantly increased in Aβ-positive mild cognitive impairment (MCI) and AD patients. The results suggest that cMD is an early marker of neuronal damage since it is observed simultaneously with Aβ deposition and is correlated with cortical thickness only in subjects without Aβ deposition. cMD changes may be driven by Aβ but not tau suggesting that direct Aβ toxicity or associated inflammation causes damage to neurons. Overall, it has some value, but there are many issues to address before it can be published.
Some comments:
1. Why are the numbers of samples in the three groups inconsistent? CN (n=21),MCI(n=50),AD (n=16).
2. CN, MCI, and AD, what are the classification criteria for the three groups?
3. A longitudinal study would provide valuable insights into the progression of cMD changes. If some longitudinal research data could be added, it might be more illustrative of the issue.
4. The image quality of Figure 2 and Figure 3 is very poor; please replace it with a clearer image.
5. The references are too outdated. Most of the references are from before 2020, please update the references.
Reviewer 2 Report
Comments and Suggestions for Authors
This pilot study is very original and novel, highlighting the potential of cMD as an early biomarker for AD. The authors made two significant discoveries in this pilot study: 1) the association of cortical diffusivity with amyloid-beta deposition, and 2) the association of cortical diffusivity with cortical atrophy in the amyloid-beta-negative subjects.
However, the authors can do more to reach a more accurate conclusion and perform a much higher-quality presentation.
1. Major concerns:
1.1. The authors did a great job of categorizing the patients based on the status of amyloid-beta. Therefore, the correlation between cortical diffusivity and amyloid-beta deposition is scientifically sound. However, the authors conclude a negative relationship between cortical diffusivity and tau aggregation without rigorously categorizing the patients based on the status of tau.
1.2. The authors used the word "driven" in the title to conclude the relationship between cortical diffusivity and amyloid-beta deposition, which is too strong to be used to integrate the results accurately. Here, "associated" should be a more accurate word in the title. Moreover, the authors did not highlight the relationship between cortical diffusivity and cortical atrophy but instead highlighted the much less rigorous relationship between cortical diffusivity and tau aggregation.
1.3. The authors should have a much more detailed statement on how the MMSE scores are measured and the relationship between the scores and mental status to expand this paper's reader pool to people unfamiliar with MMSE scores.
1.4. Table 1 has a great and accurate layout. However, the parameter label in Table 2 is confusing.
1.5. The layout of panels A, B, and C of Figure 2 does not match Table 2. Please replot it in a format like panel D in Figure 2. It would also be better if the authors could add the statistical conclusion to each panel in Figure 2.
1.6. Figure 3 is the best way to present the results of this study. The authors should plot the correlation between cMD and amyloid-beta deposition in a format similar to Figure 3 and use Spearman's rank correlation to evaluate the results. The authors should also use the same approach to assess the relationship between cMD and tau deposition.
2. Minor:
The sentence from line 184 to line 186 is grammatically inaccurate. The authors may change it to "Individual representations of cMD, Aβ and tau deposition, and cortical thickness are illustrated in Figure 2 for the frontal cortex (Figure 2A), parietal cortex (Figure 2B), temporal cortex (Figure 2C), and regional cortical thickness (Figure 2D)."
Reviewer 3 Report
Comments and Suggestions for Authors
Dear Authors
Congratulation for conducting his study, which suggests that cortical mean diffusivity (cMD) as a promising early biomarker for AD, and in detecting microstructural changes. cMD reflects neuronal damage linked to Aβ deposition, with no correlation to cortical thickness in Aβ-positive patients, indicating different underlying mechanisms. However well designed and written:
1. I would suggest to rephrase the study title, as TAU is a hallmark pathology of the disease at a later stage; it could be misleading to readers.
2. What are the underlying biological mechanisms for cortical thickness in patients with AB negative to support your findings?
3. What is the threshold to distinguish between regular age-related changes of cortical atrophy in AD or other neurodegenerative diseases?
4. Please improve the demographic parameters to study the othe exposure and comorbidity for the study participants to see the real impact of cMDs in AD, such as hypertension, diabetes, Cholesterolemia, etc.?
5. What is the rationale to have education as parameter in demographic table, please explain in the discussion or mathodology.
6. It seems very early to say the microstructural changes accurs due to only Aβ-toxicity by a cross sectional study. At least a prospective cohort study with repetitive imaging and blood parameters are necessary to prove the hypothesis when you are investigating a biomarker. I hope this pilot study will have positive effect in conducting a longitudinal cohort with stratified groups.
Round 2
Reviewer 1 Report
Comments and Suggestions for Authors
The authors has made detailed revisions, and I agree to publish it.
Reviewer 2 Report
Comments and Suggestions for Authors
The revised version of this manuscript is much better.